# Th17-Dependent Nasal Hyperresponsiveness Is Mitigated by Steroid Treatment

**DOI:** 10.3390/biom12050674

**Published:** 2022-05-06

**Authors:** Shusaku Ueda, Kento Miura, Hideki Kawasaki, Sawako Ogata, Norimasa Yamasaki, Shuka Miura, Akio Mori, Osamu Kaminuma

**Affiliations:** 1Department of Disease Model, Research Institute of Radiation Biology and Medicine, Hiroshima University, Hiroshima 734-8553, Japan; b185219@hiroshima-u.ac.jp (S.U.); kmiura@hiroshima-u.ac.jp (K.M.); b183516@hiroshima-u.ac.jp (H.K.); ogata3@hiroshima-u.ac.jp (S.O.); hlnkrcf@hiroshima-u.ac.jp (N.Y.); shu0314@hiroshima-u.ac.jp (S.M.); 2Clinical Research Center for Allergy and Rheumatology, National Hospital Organization, Sagamihara National Hospital, Kanagawa 252-0392, Japan; mori-kkr@umin.ac.jp

**Keywords:** inflammation, nasal hyperresponsiveness, Th17 cell, dexamethasone, mouse

## Abstract

Th17 cells are implicated in allergic inflammatory diseases, including allergic rhinitis (AR), though the effect of steroids on Th17 cell-dependent nasal responses is unclear. Herein, we investigated a nasal inflammation model elicited by allergen provocation in mice infused with Th17 cells and its responsiveness against steroid treatment. We transferred BALB/c mice with Th17 cells, which were differentiated in vitro and showed a specific reaction to ovalbumin (OVA). We challenged the transferred mice by intranasal injection of OVA and to some of them, administered dexamethasone (Dex) subcutaneously in advance. Then, we assessed immediate nasal response (INR), nasal hyperresponsiveness (NHR), and inflammatory cell infiltration into the nasal mucosa. The significant nasal inflammatory responses with massive neutrophil accumulation, INR, and NHR were induced upon allergen challenge. Allergen-induced INR and NHR were significantly suppressed by Dex treatment. This study suggested the effectiveness of steroids on Th17 cell-mediated nasal responses in AR.

## 1. Introduction

In allergic rhinitis (AR) patients, exposure to a specific allergen induces a variety of nasal symptoms, represented by sneezing, rhinorrhea, and congestion. The series of immediate nasal response (INR) is often accompanied by inflammatory responses, elicited in the nasal mucosa [1]. In particular, the infiltration of inflammatory cells and development of nasal hyperresponsiveness (NHR), which causes elevated nasal symptoms to allergens, as well as non-specific stimuli, are observed [2].

Although the mechanism of allergic nasal inflammation is not fully understood, it has been pointed out that CD4^+^ T cells act as a key player. An increase in CD4^+^ T cells that synthesize critical cytokines is detected in the nasal mucosa and submucosa of AR patients [3]. Nasal eosinophil accumulation and NHR elicited by allergen challenge in sensitized mice are mitigated by depletion of CD4^+^ T cells [4]. Allergen-induced INR and NHR are induced in wild-type mice by adoptive transplantation of several subsets of allergen-reactive helper T (Th) cells, e.g., Th2, Th9, and Th17 cells [4,5]. Th2 and Th9 cells induce eosinophil-dominated nasal inflammation, while Th17 cell-mediated inflammation is characterized by massive accumulation of neutrophils [4].

Steroids are mainly used for the treatment of AR, though the existence of patients who are resistant to steroid-based therapy is a serious problem [6]. Th17 cells have been involved in developing steroid-resistant allergic inflammation. It is known that, in mice transferred with Th17 cells, allergen-induced infiltration of neutrophils in the lungs and bronchial hyperresponsiveness (BHR) were not effectively suppressed by steroid administration [7]. However, whether steroids affect Th17-dependent nasal responses has not been evaluated.

We, here, investigated the efficacy of dexamethasone (Dex) on allergen-evoked INR, NHR, and inflammatory cell infiltration in the nasal cavity in mice infused with Th17 cells. We also examined the effect on Th17-related gene expression in the nasal-associated lymphoid tissue (NALT).

## 2. Materials and Methods

### 2.1. In Vitro Development of Allergen-Reactive Th17 Cells

Allergen-reactive Th17 cells were developed as previously described [4,8]. In brief, ovalbumin (OVA)-reactive CD4^+^ T cells were prepared from spleen cells of BALB/c background transgenic mice, DO11.10/RAG2^−/−^, by magnetic cell sorting with an EasySep Mouse CD4^+^ T Cell Isolation Kit (Veritas, Santa Clara, CA, USA). The cells were co-cultured in the presence of X-ray-irradiated spleen cells in AIM-V medium (Thermo Fisher Scientific, Waltham, MA, USA) with 10% fetal calf serum. At the beginning of culture, we added 0.3 μM OVA323-339 synthetic peptide (Scrum Inc., Tokyo, Japan), 10 ng/mL human IL-1β (PeproTech, Rocky Hill, NJ, USA), 20 U/mL IL-2 (PeproTech), 20 ng/mL IL-6 (PeproTech), 10 ng/mL IL-23 (R & D Systems, Minneapolis, MN, USA), 1 ng/mL human TGF-β (BioLegend, San Diego, CA, USA), 10 ng/mL TNF-α (PeproTech), 10 μg/mL anti-IL-4 (Abcam, Cambridge, UK), and 10 μg/mL anti-IFN-γ (R4-6A2, eBioscience, San Diego, CA, USA). Cells were collected following seven-day culture and used for the transfer. Successful polarization of Th17 cells has been reported elsewhere [4], and was confirmed by comparing *Il4* and *Il17a*-expressing activity with that of Th2 cells (Appendix A).

### 2.2. Allergen-Induced Nasal Responses

Each seven- to eight-week-old female BALB/c mouse (Charles River Laboratories Japan, Kanagawa, Japan) was intravenously injected with the polarized Th17 cells (2 × 10^7^) on day 0 (Figure 1). Six hours later, these mice received daily injection of 20 μL OVA (30 mg/mL; Sigma, St. Louis, MO, USA) or phosphate-buffered saline (PBS) through the intranasal route on days 0–4 (Figure 1). INR was evaluated on day 3 by the count of sneezes measured for 5 min immediately following the OVA challenge. NHR was evaluated 6 h following the final OVA injection on day 4 by the count of sneezes measured for 5 min immediately after the injection of 10 μL histamine (100 mM; Nacalai tesque, Kyoto, Japan) [4]. Nasal lavage fluid (NALF) and NALT were subsequently collected. We classified inflammatory cells recovered in the NALF by referring the morphological criteria as described previously [4]. This procedure did not elicit any inflammatory features in the lungs [9]. Nasal OVA challenge to mice without allergen-specific Th cell transfer did not induce any inflammatory response [4]. In some assessments, mice were injected subcutaneously with 2.5 or 10 mg/kg Dex (Tokyo Kasei, Tokyo, Japan) which was suspended in PBS containing 0.5% Tween-20 (Sigma) 30 min prior to each allergen challenge on days 0–3. To evaluate the effect on NHR developmental process but not on direct influence on histamine-induced sneezing response, Dex was not administered on day 4. Although steroids are normally administered in the nose of AR patients, our procedure enabled us to compare the pharmacological effect of Dex with that we previously obtained in Th2 cell-transferred mice [10], without concerning its distribution following administration. Vehicle alone displayed no effect on any experimental parameter evaluated in this study.

### 2.3. Gene Expression Assessment

Following the extraction of total RNA from the NALT, reverse transcription was performed using random primers with SuperScript VILO cDNA Synthesis Kit (Thermo Fisher Scientific), and then quantitative RT-PCR for *Rorc* (Mm01261022_m1) and *Il17a* (Mm00439618_m1) was carried out using Taqman gene expression probes (Thermo Fisher Scientific) on the ABI StepOnePlus^TM^ Real-Time PCR System (Thermo Fisher Scientific). The relative transcript levels were normalized to *Gapdh* (4351309) expression as an endogenous reference.

### 2.4. Statistical Analysis

Data are displayed as the arithmetic median with interquartile range. Data analyses were conducted using the graphics and statistics program Prism v9.3.1 (GraphPad Software, San Diego, CA, USA). In some experiments, Dex-responder and -non-responder mice were divided based on the NHR values less and more than the median of Dex-treated mice, respectively. Statistical analyses were conducted by one-way analysis of variance and additional Dunnett’s test. *p* < 0.05 was recognized to demonstrate statistical significance.

## 3. Results

For evaluating the ability of Th17 cells to mediate allergic nasal inflammation and its steroid responsiveness, allergen-reactive Th17 cells were prepared from CD4^+^ T cells in DO11.10/RAG2^−/−^ mice expressing T cell receptor, corresponding to OVA. After the confirmation that Th17 cells were differentiated adequately by assessing cytokine production [8], the cells were infused to BALB/c mice. Repeated challenge with intranasal OVA injection (Figure 1) significantly induced INR in those mice (Figure 2). We have elucidated that INR developed in mice transferred with various Th subsets is caused mainly by reflecting NHR to non-specific stimuli but not IgE-mediated mast cell degranulation [4]. As a result, we observed significant NHR (Figure 2) and massive and weak infiltration of neutrophils and eosinophils, respectively, in the nasal mucosa (Figure 3) after allergen exposure in Th17 cell-infused mice. OVA injection affected the infiltration of neither lymphocytes nor macrophages significantly.

It was previously reported that Th17 cell-mediated allergic bronchial inflammation was not affected by steroid treatment [7]. Therefore, we investigated the effect of Dex treatment on Th17 cell-dependent nasal responses. Allergen-evoked INR and NHR in Th17 cell-infused mice were significantly mitigated by administrating 2.5 and/or 10 mg/kg Dex (Figure 2). Neutrophil migration was also suppressed by Dex, though the effect was not statistically significant (Figure 3). The relative weak eosinophil accumulation was significantly alleviated by 2.5 mg/kg Dex (Figure 3).

Next, we evaluated the expression of Th17-characteristic genes, *Rorc* and *Il17a*, by quantitative RT-PCR (Figure 4). Exposure of Th17 cell-transferred mice to the allergen increased the expression of *Rorc* and *Il17a* in the NALT, though the responses were not statistically significant. Dex administration, particularly at 2.5 mg/kg, tended to inhibit the expression of *Rorc* (60% of the OVA-challenged control, *p* = 0.81) but not *Il17a*. Consistently, *Rorc* expression in OVA-challenged and/or Dex-administered mice was positively but weakly correlated with NHR and nasal accumulation of total cells, neutrophils and eosinophils (Appendix A). Considering the existence of Dex-responder and -non-responder animals, a relatively stronger correlation was observed between *Rorc* expression and NHR in the non-responder group (Appendix A).

## 4. Discussion

We showed that Th17 cells induce neutrophilic inflammation with marked hyperresponsiveness, in both the lung and nasal mucosa [4,8]. In the present study, Dex showed an inhibitory effect on Th17 cell-mediated nasal inflammation, particularly on INR and NHR. The present results contradict the previous report, showing that Dex was ineffective against allergen-induced BHR in Th17 cell-transferred mice [7]. We also found that Th9 cell-mediated NHR, but not BHR, was suppressed by Dex [5,11], and that both NHR and BHR developed in Th2 cell-transferred mice were attenuated [10,11]. The difference in the effect of Dex on those models is hard to explain by the effect on each Th subset function, because Dex suppresses subset-specific cytokine synthesis in Th2 and Th9 cells, but not Th17 cells [7,11]. As some genes or cytokines are involved in both Th9 and Th17 differentiation [12], these subsets might produce common factors responsible for steroid resistance, at least in the induction of bronchial inflammation. In addition, the doses of Dex that effectively inhibited INR, NHR, and cellular infiltration differed (Figure 2 and Figure 3), suggesting the existence of appropriate steroid doses for inhibiting each allergic inflammatory response in each target tissue. Since the tissue-specific surrounding environment, such as chemokines, adhesion molecules, and allergen-presenting cells, probably determine the Th17 cell-mediated steroid responsiveness in each tissue [13], it is further needed to detect steroid-sensitive factors involved in the different activation processes of Th17 cells in the individual target tissues.

Retinoic-acid-receptor-related orphan receptor γt, an isoform of transcription factors coded in the *Rorc* gene [14], is crucial for the development of Th17 cells and transcription of Th17-specific cytokines, IL-17A and IL-17F [15]. IL-17A is involved in Th17-mediated allergic inflammation [16]. Elevated IL-17A expression was detected in the airway biopsy specimens and sputa from patients with moderate-to-severe asthma [17,18]. IL-17A directly raised the contractility of airway smooth muscles in humans and mice [19]. Administration of an anti-IL-17A antibody mitigated airway inflammation with hyperresponsiveness elicited by house dust mites in mice [20].

In this study, the expression of *Rorc* and *Il17a* in the NALT of Th17 cell-infused mice was slightly augmented by the allergen challenge, suggesting the de novo tissue infiltration of Th17 cells. However, the tendency towards the suppression of *Rorc* but not *Il17a* expression was observed by Dex treatment. Therefore, down-regulation of Th17 cell migration may be involved in the inhibitory effect of Dex on the nasal inflammation. Moreover, IL-17A-synthesizing activity in less-infiltrated Th17 cells seemed to be enhanced at the single-cell level. It is consistent with the previous in vitro study demonstrating the enhancement of IL-17A production from T cells of steroid-resistant asthma patients by Dex [21]. The effect on *Il17a* expression can be excluded, at least in our experimental conditions, from the suppression of INR and NHR by Dex. *Rorc* expression was weakly correlated with NHR and cellular infiltration, suggesting the contribution of Th17 cells to allergen-induced nasal inflammation through IL-17A-independent mechanisms. The relatively stronger correlation observed between *Rorc* expression and NHR in the Dex-non-responder group may provide hints to develop a novel means to treat steroid-resistant patients in several allergic diseases [22].

In summary, these results indicate that Th17 cell-mediated allergic inflammation in the nasal mucosa, especially INR and NHR, is suppressed by Dex treatment. It is suggested that it is possible to manage AR by targeting Th17-mediated nasal responses.

## Figures and Tables

**Figure 1 biomolecules-12-00674-f001:**
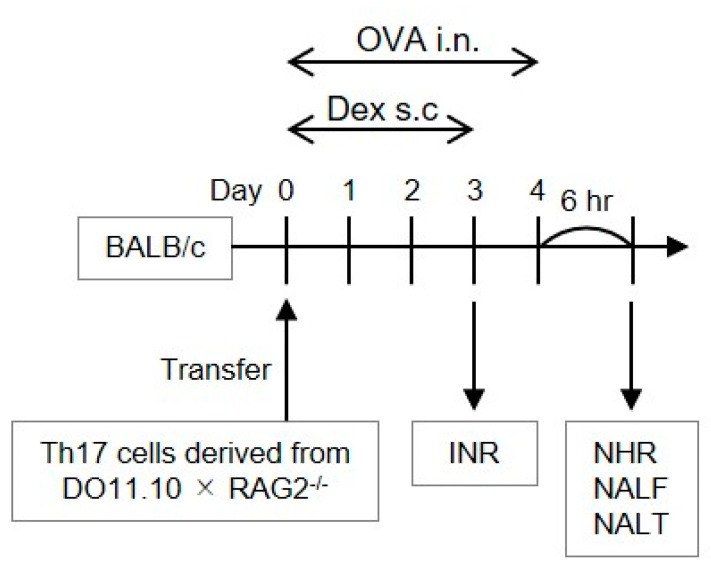
Procedural overview of the experiments. BALB/c mice infused with Th17 cells differentiated in vitro were challenged by intranasal (i.n.) ovalbumin (OVA, 600 μg/head) injection once daily on days 0–4. Before challenge on days 0–3, they received dexamethasone (Dex, 2.5 or 10 mg/kg) through a subcutaneous route. Immediately after the OVA injection on day 3, immediate nasal response (INR) was assessed. Six hours following the last OVA injection on day 4, nasal hyperresponsiveness (NHR), nasal lavage fluid (NALF), and nasal-associated lymphoid tissue (NALT) were assessed.

**Figure 2 biomolecules-12-00674-f002:**
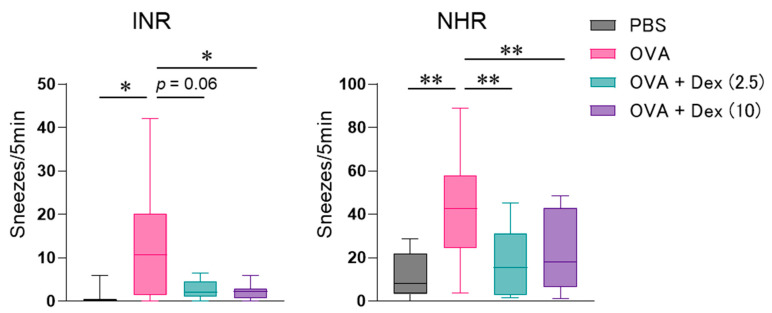
Effect of dexamethasone (Dex, 2.5 or 10 mg/kg) on allergen-induced immediate nasal response (INR) and nasal hyperresponsiveness (NHR) in Th17 cell-infused mice. INR was assessed by the count of sneezes measured for 5 min immediately after ovalbumin (OVA) challenge on day 3. NHR was assessed by the count of histamine-induced sneezes 6 h following the final OVA injection on day 4. Data are displayed as the box plot of 5–14 mice. PBS, phosphate-buffered saline. * *p* < 0.05, and ** *p* < 0.01, compared with OVA-injected mice (pink bar).

**Figure 3 biomolecules-12-00674-f003:**
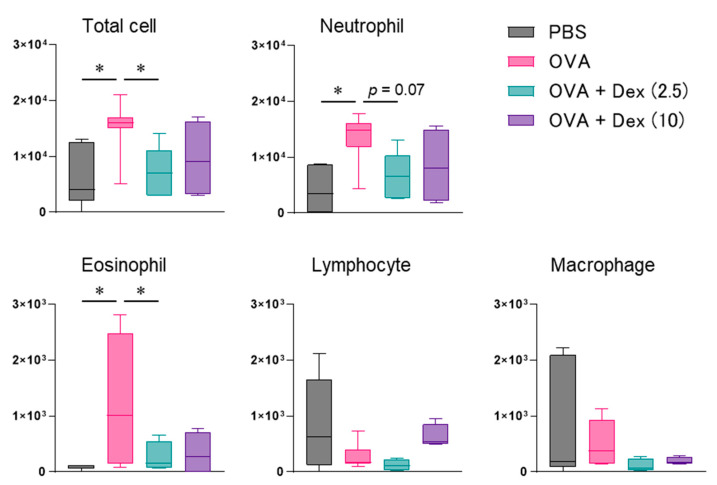
Effect of dexamethasone (Dex, 2.5 or 10 mg/kg) on allergen-evoked nasal cell infiltration in Th17 cell-infused mice. Inflammatory cell numbers in nasal lavage fluid (NALF) were measured 6 h following the final ovalbumin (OVA) injection. Data are displayed as the box plot of 4–7 mice. PBS, phosphate-buffered saline. * *p* < 0.05, compared with OVA-injected mice (pink bar).

**Figure 4 biomolecules-12-00674-f004:**
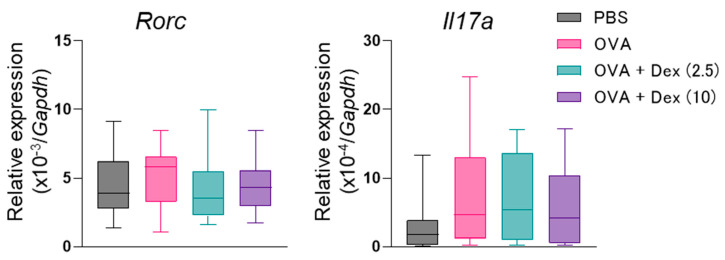
Effect of dexamethasone (Dex, 2.5 or 10 mg/kg) on gene expression in the nasal-associated lymphoid tissue (NALT) of Th17 cell-transferred mice. Expression of *Rorc* and *Il17a* mRNA in the NALT was evaluated. Data are displayed as the box plot of 6–11 mice. PBS, phosphate-buffered saline; OVA, ovalbumin.

## Data Availability

The data presented in this study are available in the manuscript and Appendix A.

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
