# Peer review of "Th17-Dependent Nasal Hyperresponsiveness Is Mitigated by Steroid Treatment"

_biomolecules, 2022, doi:10.3390/biom12050674_

Round 1
Reviewer 1 Report
The study is interesting in the field of Th17-dependent Nasal Hyperresponsiveness in mice. However, there are some issues which should be done to improve the manuscript:
- Methods: It is more relevant if one group of mice treated with local corticosteroid could be done in this study. In reality, nasal hyperresponsiveness might be treated by local treatment with corticosteroids.
- Results: The authors should not give their comments in the results section. The main results should be done in this section.
Reviewer 2 Report
Despite really interesting study on the effects of steroids in mice model of allergic rhinitis transferred with Th17 cells, some questions and comments require the Authors response.
- Could the Authors provide more precise citation for the first sentence of the introduction third paragraph (line 42-43)? The reference used currently is corresponding more to the asthma and not allergic rhinitis. Moreover, there is an information on serious problem of steroid-resistant patients, which might be true for asthma (with around 5-25% of patients). However, it might be essential at that point focus more on allergic rhinitis and its relation to steroids and possible drawbacks.
- It would be beneficial for the manuscript to provide at least one graphical element demonstrating effectiveness of inducing Th17 cell to support the model reliability.
- Is it possible by any chance that Th17 cells from RAG2-/- mice could induce immune response of the BALB/c mice in context of histocompatibility?
- Apart from confirmation of the Th17 generation, histological presentation of the nasal area after allergic rhinitis induction would be essential. Moreover, the results of treatment applied should also be supported by sample representative graphs. That corresponds also to the results section and the data presented there.
- The Authors demonstrated the gene expression analysis data as normalized to GAPDH. Considering presence of the vehicle group it might probably be better to further calculate the data as 2^(delta-delta)ct to visualize expression change in comparison to the vehicle, or even OVA challenged untreated mice.
- In most subsections of the materials and methods there are no information on the devices used inter alia for RT-PCR assessment and software implemented in biostatistical analysis. In addition, the whole section on histological examination of the nasal samples is missing.
- Considering the fact that Th17 might affect responses to the steroids, would it not be beneficial for the study to include groups without that population of Th cells? That could allow to compare the nasal tissue response to the steroids in presence or absence of the IL-17-producing cells.
- Did the Authors evaluated the differences in response to different concentrations of the steroid? In Figure 3 there seems to be a huge difference between two doses used, however, no significance was achieved.
- Regarding the results presented in the Figure 4, what were the exact p-values of the differences as within the text there is a suggestion of ‘tendencies’ is some cases? Despite there is a limitation in mice in each group, did the Authors considered using stratification based on the INR, NHR and other hyperreactivity-related parameters and then demonstrate Th17-related genes in those subgroups?
- Was the correlation performed between Th17-related and other allergic reaction parameters within studied groups?
- Some parts of the discussion section seems to miss references, like for example last two sentences of the first paragraph.
- The conclusion within third paragraph of the discussion should be verified. In the results section there were no significant differences in Th17-related genes in tested groups. Even indication of tendency would require certain level of the p-value.
Reviewer 3 Report
1. In the OVA model, why the time of Dex intervention is one day less than OVA, and whether this will affect the efficiency. Would it be more realistic to induce INR and NHR in OVA first and then start Dex intervention? I would like to know the rationale for such a design, or if there is any literature that I can refer to in the past.
2. The benefits of different drug doses of Dex are significant in both INR and NHR responses, but the response of immune cells is inconsistent, but the suppressive effect of 2.5 is better.
The number of lymphocyte and Marcophage in the group with OVA intervention was lower than that in the control group, can the reason be explained in the "Results or Discussion".
The text description of the results is a bit too simple.
In Figure 4, Dex has no significant effect on the II17a gene, but has an inhibitory effect on the inflammatory response.
6. Is there any other relevant data that can be presented to clarify the main mechanism?
Reviewer 4 Report
The reviewed and evaluated article is a revised and re-submitted version of an earlier work. Probably it is corrected according to previous suggestions. I compared the two versions and I conclude that, in fact, the changes are introduced in the illustrative material (charts) and in the Discussion part of the results, and there are clearly noticeable and have a beneficial effect. These corrections had a very good impact on the substantive side of the work and on the clarity of the arguments contained in the discussion. The other parts of the work remained unchanged significantly, so I think they were positively evaluated previously. I fully agree with this assessment.
This article includes relatively little experimental material and perhaps that is why it was not previously approved. However, it should be considered that it is presented as a Communication. In my opinion, it meets all the formal criteria set for scientific communications. It also meets the criteria of novelty and originality of the research presented and the correctness of the description and discussion of the research performed.
I recommend accepting it for publication as a Communication, without further correction.
Round 2
Reviewer 1 Report
The manuscript can be published in its revised version because the authors have responded all the reviewer' s comments appropriately.
Reviewer 3 Report
I have no other comments.